# Factors influencing career preference in mental health among nursing students and intern nurses in Dar es Salaam, Tanzania

**Sofia Samson Sanga[1], Edith A. M. Tarimo[2], Joel Seme Ambikile[1]***

**1** Department of Clinical Nursing, Muhimbili University of Health & Allied Sciences (MUHAS), Dar es Salaam, Tanzania, **2** Department of Nursing Management, Muhimbili University of Health & Allied Sciences (MUHAS), Dar es Salaam, Tanzania

* joelambikile@yahoo.com

**Data Availability Statement:** All data are fully available without restriction in the paper.

**Funding:** SSS received funding for the study. The study was supported by the Government of

## Abstract

Worldwide, the prevalence of mental health, neurological, and substance use (MNS) disorders has been on the rise and remains a significant leading cause of disease burden. Sub-Saharan Africa (SSA) shares a fair burden of MNS with depressive disorders being the most prevalent in this region. A huge treatment gap for MNS exists, with lack of appropriate human resources and expertise for service delivery being one of the key barriers. Pre-service and in-service training plays a vital role in developing human resource for mental health. However, low or lack of career interests in mental health has been documented among students. A cross-sectional study was conducted between April and May 2021 to determine factors influencing career preference in mental health among nursing students and intern nurses at Muhimbili University of Health and Allied Sciences (MUHAS) and Muhimbili National Hospital (MNH) respectively in Dar es Salaam, Tanzania. Sixty-eight (68) nursing students at MUHAS who had covered the mental health nursing course and 83 intern nurses who had rotated at the MNH Psychiatry and Mental Health department participated in the study using consecutive sampling. A pre-tested structured self-administered questionnaire was used to collect data, followed by analysis with version 25 of the Statistical Package for the Social Sciences. The Chi-square test and logistic regression were performed to determine factors associated with career preference. One third (33.1%; n = 50) of participants had career preference in mental health nursing. Living with a person with mental illness (adjusted odds ratio [AOR]: 4.350; 95% CI: 1.958, 9.664; $p$ <0.001), awareness of possible career advancement in mental health (AOR: 16.193; 95% CI: 2.022, 129.653; $p$ = 0.009), awareness of possible income generation in mental health career (AOR: 6.783; 95% CI: 2.295, 20.047; $p$ = 0.001), and satisfaction with psychiatric working environment (AOR: 6.753; 95% CI: 2.900, 15.726; $p$ <0.001), were significantly associated with career preference in mental health. Low mental health career preference among university nursing students and intern nurses jeopardizes the future of the mental health nursing profession and may complicate the already existing shortage of human resource for mental health. The higher learning institutions, health facilities, and the Ministry of Health may need to take deliberate actions to ensure that interest to pursue a career in mental health is built among

Tanzania through the Ministry of Health which can be found at https://www.moh.go.tz/. The funder had no role in study design, data collection and analysis, decision to publish, or preparation of the manuscript.

**Competing interests:** The authors have declared that no competing interests exist.

students and interns. Further research is needed to provide more insight into how the psychiatric working environment affects career preference in mental health.

## Introduction

The prevalence of mental health, neurological, and substance use (MNS) disorders has been on the rise and remains a significant leading cause of disease burden worldwide [1]. In 2016, mental and addictive disorders affected more than one billion people globally, causing 7% of all global burden of disease as measured in DALYs and 19% of all years lived with disability [2]. Sub-Saharan Africa (SSA) shares a fair burden of mental disorders with depressive disorders being the most prevalent in the region [3, 4]. In resource-poor countries such as SSA, there is a huge treatment gap i.e. large proportion (estimated to be between 76 and 85%) of patients with MNS disorders do not receive appropriate treatment for their condition [5–7].

The treatment gap for MNS disorders in low- and middle-income countries (LMICs) has been attributed to various factors. Lack of appropriate human resources and expertise for service delivery is one of the key barriers that contributes to this gap, thus hindering delivery of essential mental health interventions [8, 9]. To address this barrier, amplified training for all cadres of mental health professionals and specialists is deemed vital [10]. Skilled human resources for mental health should be made available at the different levels of service delivery. Pre-service and in-service training of healthcare workers on mental health issues is an essential prerequisite for delivery of mental health services in various platforms [11]. However, low or lack of career interests in mental health has been documented among students studying in health-related training institutions which affects the availability of healthcare workers in mental health and psychiatric settings [12, 13].

Studies have revealed various factors influencing career interest or preference in mental health among students. Students who spend more time (more than a month) doing their psychiatry ward rotation are more likely to develop interest in mental health than those who spend less time (not more than a month or no psychiatry rotation) [12]. A greater knowledge of mental health, better perception of job prospects in psychiatric nursing than other fields, and having a positive attitude towards psychiatry are also linked to choosing a career in mental health nursing [14, 15]. Apart from the direct influence of stigma towards people with mental illness, the exposure of students considering a career in mental health to stigmatizing comments from family members, friends, and the general public on their career choice, negatively affects their decision to pursue their career [16]. Moreover, while having a mental health condition and a family history of psychiatric illness are associated with a greater likelihood of considering a career in mental health, the difficult working environment in psychiatry is associated with a less career interest in this area [12, 14, 17].

The increasing number of people with MNS disorders implies that more healthcare providers with proper training in mental health are needed [2]. Nurses, who constitute the largest health workforce and a core aspect of strengthening essential health services in the front line, are among the potential healthcare professionals to be considered for career advancement in mental health [18]. However, mental health and psychiatric nursing is often not the preferred option for nursing students [12]. A number of studies have shown that many students do not prefer choosing mental health as a career [12, 13]. This leads to shortage of staff and it is one of the major reasons the mental health services continue to suffer [19]. Tanzania is one the countries largely affected by shortage of human resource for mental health [20]. This study was

conducted to determine factors influencing career preference in mental health among nursing students and intern nurses in Dar es Salaam, Tanzania.

This study adapts the conceptual framework from a study conducted in Ghana to determine factors that influenced Community Mental Health Workers to choose careers in mental health [21]. The factors influencing career preference in mental health nursing in this adapted conceptual framework are categorized in three groups. The first category are individual factors such as intrinsic motivation, pre-service education, experience of having family member with mental challenges, and a sense of professional identity. The second category are institutional factors including the availability of benefits (e.g. salary, transport, and housing), opportunity for career advancement, working conditions (e.g. availability of supplies, clarity of roles, and control over practice), relationship with other health workers, and adequate staffing. The third category are community factors consisting of community acceptance, stigma of working in mental health, and perception about risk of working in mental health.

## Materials and methods

### Study design

This was a cross-sectional study conducted at Muhimbili University of Health and Allied Sciences (MUHAS) and Muhimbili National Hospital (MNH) in Dar es Salaam, Tanzania between April and May 2021. MNH serves as a tertiary and national referral and teaching hospital, with 1500 bed capacity and attending 1000 to 2000 outpatients and 1000 to 1200 inpatients per week. The hospital has about 8 departments including the Psychiatry and Mental Health. MUHAS is a premier public university in Tanzania is playing a leading role in training health professionals at different levels (diploma, undergraduate, and postgraduate). The university has 5 schools (including nursing), nine directorates, and two institutes.

### Study population

This study involved diploma nursing students at the Institute of Allied Sciences–MUHAS (those who were undergoing 3-years studies to acquire the National Technical Award level 6), undergraduate nursing students at the School of Nursing–MUHAS (those undertaking a 4-year bachelor degree in nursing), and intern nurses at MNH (those who had a bachelor degree in nursing programs and were undergoing a one-year rotation in various MNH departments).

### Inclusion and exclusion criteria

The inclusion criteria were all nursing students who had covered the mental health nursing course (both theory and practice) and all intern nurses who were or had completed rotating at the MNH Psychiatry and Mental Health department. At the time of data collection, MUHAS had a total of 40 diploma and 49 undergraduate nursing students, and MNH had 80 intern nurses who had fulfilled the criteria. Those who were not available during the period of data collection for various reasons (e.g. on holiday, attending other duties, and sick) were excluded from the study.

### Data collection method and procedures

A self-administered questionnaire was employed to collect data from participants, i.e., nursing students who were present in a class and intern nurses who were present at work in all MNH departments on the days of data collection. The questionnaire had three parts consisting of questions on socio-demographic characteristics, career preference in mental health and related factors, and attitude towards mental health and mental illness. The socio-demographic

characteristics included age, sex, marital status, level of education, and religion. Career preference was assessed by the question "Would you prefer to develop career in mental health nursing" with a 'Yes' or 'No' response. The proportion of participants with a career preference in mental health was calculated by using a percentage score of those who provided a "Yes' response. Questions on factors related to career preference included experience of living with a person with mental illness, history of being diagnosed or treated for mental illness, awareness of possible career advancement in mental health, awareness of possible income generation in mental health career, and satisfaction with psychiatric working environment. These factors were scored either by a 'Yes' or 'No' response, or by 'Agree' or 'Disagree' response. Attitude questions were adapted from a previously validated Attitudes to Mental Illness Questionnaire (AMI) [22] and used a Likert scale with 4 options (strongly agree, agree, disagree, and strongly disagree) which were scored 4, 3, 2, and 1 respectively. The highest and lowest possible attitude scores were thus 28 and 1 respectively, and a score of 1 to 14 and 15 to 28 were used to assign negative and positive attitude, respectively. Part of the tool measuring attitude was tested for its internal consistency reliability and the Cronbach's alpha was 0.779. The tool was piloted among 6 nursing students (3 undergraduates and 3 diploma students) whose data were excluded from the analysis. No questions were modified in the tool following the pilot test as they were all clear.

The authors collected data by visiting nursing students and intern nurses in their classes and departments, respectively. Students were approached immediately after they had finished classes and before they went for break while intern nurses were contacted through their nurse in-charges in the wards or units. The purpose of the study was explained to the potential respondents followed by obtaining a written informed consent. The respondents were provided with the questionnaire to be filled during the break time (for students) or at the end of the shift (for intern nurses). The first author collected all the filled questionnaires immediately after the exercise and checked for completeness. The numbers of respondents who filled the questionnaire and included in the final analysis are shown in Table 1.

## Data analysis

The Statistical Package for the Social Sciences (SPSS) for windows version 25 was used to analyze the data. Through univariate analysis, descriptive statistics were summarized and interpreted using tables, means, standard deviation, frequencies, and percentages. A Chi-square Test (or Fisher's Exact Test) was performed to assess associations between variables. The logistic regression was used to assess factors associated with career preference in mental health nursing. Spearman's rank correlation coefficient was used to measure the association between awareness of possible career advancement in mental health and awareness of possible income generation in the specialty. A $p$ value of $< 0.05$ was used to determine statistical significance.

## Ethics approval

The ethical approval was obtained from the Muhimbili University of Health and Allied Sciences (MUHAS) Research and Ethics committee, with Ref. No. DA.282/298/01.C/. Permission

**Table 1. Numbers of study participants and subjects included in the final analysis.**

| Institution | Nature of Subjects | Eligible Participants | Subjects included in the final analysis | % of subjects included in the final analysis |
| --- | --- | --- | --- | --- |
| MUHAS | Diploma students | 40 | 34 | 85 |
| MUHAS | Undergraduate students | 49 | 49 | 100 |
| MNH | Intern nurses | 80 | 68 | 85 |
| Total | | 169 | 151 | 89 |

to conduct the study was obtained from the Dean of the School of Nursing–MUHAS and the Coordinator of Teaching, Research, and Consultancy Unit–Muhimbili National Hospital on behalf of the Executive Director. The potential respondents were informed of their role in the study and the expectations were clarified. They were identified by numbers for anonymity and a written informed consent was obtained from each respondent before commencement of data collection. Respondents were also informed of their right to withdrawal from the study at any time if they wished to do so for whatever reasons, and that this decision would not affect their entitlement.

## Results

### Socio-demographic characteristics

Of the 151 respondents, the proportions of males and females were almost equal (50.3% vs. 49.7%). The mean age (SD) was 25.7 (3.4) and majority were single (70.2%). Table 2 shows respondents' socio-demographic characteristics. There was a statistically significant difference between nursing students and intern nurses with regard to having a spouse (married or cohabiting) (19.1% vs. 38.6%, Chi-square test, $p = 0.009$), ever being diagnosed with a mental illness (22.1% vs. 2.4%, Chi-square test, $p < 0.001$), and being satisfied with the psychiatric working environment (47.1% vs. 24.1%, Chi-square test, $p < 0.003$). Nursing students were more likely

**Table 2. Sociodemographic characteristics according to level of nursing professional education (N = 151).**

| | | | Level of professional education | | | | *p* value |
|---|---|---|---|---|---|---|---|
| | | Total | Student nurses | | Intern nurses | | |
| | | n(%) | n | % | n | % | |
| Mean Age (SD) | | 25.7(3.4) | 24.0(1.8) | | 27.1(3.7) | | |
| Sex | Male | 76(50.3) | 32 | 47.1 | 44 | 53.0 | 0.467 |
| | Female | 75(49.7) | 36 | 52.9 | 39 | 47.0 | |
| Spouse Status | No spouse | 106(70.2) | 55 | 80.9 | 51 | 61.4 | 0.009 |
| | Has a spouse | 45(29.8) | 13 | 19.1 | 32 | 38.6 | |
| Religion | Christian | 92(61.7) | 40 | 60.6 | 52 | 62.7 | 0.799 |
| | Muslim | 57(38.3) | 26 | 39.4 | 31 | 37.3 | |
| Career preference in MH | No | 101(66.9) | 35 | 51.5 | 66 | 79.5 | <0.001 |
| | Yes | 50(33.1) | 33 | 48.5 | 17 | 20.5 | |
| Ever lived with a person with MI | No | 94(62.3) | 38 | 55.9 | 56 | 67.5 | 0.144 |
| | Yes | 57(37.7) | 30 | 44.1 | 27 | 32.5 | |
| Ever diagnosed/treated for MI | No | 134(88.7) | 53 | 77.9 | 81 | 97.6 | <0.001 |
| | Yes | 17(11.3) | 15 | 22.1 | 2 | 2.4 | |
| Possible career advancement in MH | Disagree | 27(17.9) | 6 | 8.8 | 21 | 25.3 | 0.009 |
| | Agree | 124(82.1) | 62 | 91.2 | 62 | 74.7 | |
| Possible income generation in MH | Disagree | 51(33.8) | 12 | 17.6 | 39 | 47.0 | <0.001 |
| | Agree | 100(66.2) | 56 | 82.4 | 44 | 53.0 | |
| MNH psychiatric working environment | Unsatisfied | 99(65.6) | 36 | 52.9 | 63 | 75.9 | 0.003 |
| | Satisfied | 52(34.4) | 32 | 47.1 | 20 | 24.1 | |
| Attitude towards MH and MI | Negative | 7(4.6) | 4 | 5.9 | 3 | 3.6 | 0.701[a] |
| | Positive | 144(95.4) | 64 | 94.1 | 80 | 96.4 | |

Chi-square test

Key:

[a] Fisher's Exact Test

to have a career preference in mental health nursing (48.5% vs. 20.5%, Chi-square test, $p < 0.001$), to agree that there is a possible career advancement in mental health nursing (91.2% vs. 77.7%, Chi-square test, $p = 0.009$), and to agree that there is income generation in mental health nursing just like other health specialties (82.4% vs. 53.0%, Chi-square test, $p = <0.001$), than intern nurses.

## Career preference in mental health nursing

As shown in Table 3, one third (n = 50; 33.1%) of participants preferred to have a career in mental health nursing, with nursing students' preference being higher (48.5%) than that of intern nurses (20.5%). There were statistically significant differences in career preference by marital status (Chi-square test, $p = 0.021$), level of professional education (Chi-square test, $p < 0.001$), ever living with a person with mental illness (Chi-square test, $p < 0.001$), possible career advancement in mental health (Chi-square test, $p < 0.001$), possible income generation in mental health (Chi-square test, $p < 0.001$), and satisfaction with psychiatric working environment (Chi-square test, $p < 0.001$).

## Factors associated with career preferences in mental health

Table 4 shows six models of logistic regression performed to analyse factors associated with career preference in mental health nursing. All models included sex, marital status, level of professional education, and religion as individual profile for independent variables. Living

**Table 3. Nursing students' and intern nurses' career preference in mental health (N = 151).**

|  |  | Career preference in mental health | | | | *p* value |
|  |  | (Yes) | | (No) | | |
|  |  | n | % | n | % | |
| Sex | Male | 27 | 35.5 | 49 | 64.5 | 0.526 |
|  | Female | 23 | 30.7 | 52 | 69.3 | |
| Spouse Status | No spouse | 29 | 27.4 | 77 | 72.6 | 0.021 |
|  | Has a spouse | 21 | 46.7 | 24 | 53.3 | |
| Level of professional education | Student nurses | 33 | 48.5 | 35 | 51.5 | <0.001 |
|  | Intern nurses | 17 | 20.5 | 66 | 79.5 | |
| Religion | Christian | 30 | 32.6 | 62 | 67.4 | 0.755 |
|  | Muslim | 20 | 35.1 | 37 | 64.9 | |
| Ever lived with a person with MI | No | 19 | 20.2 | 75 | 79.8 | <0.001 |
|  | Yes | 31 | 54.4 | 26 | 45.6 | |
| Ever diagnosed/treated for MI | No | 43 | 32.1 | 91 | 67.9 | 0.453 |
|  | Yes | 7 | 41.2 | 10 | 58.8 | |
| Possible career advancement in MH | Disagree | 1 | 3.7 | 26 | 96.3 | <0.001 |
|  | Agree | 49 | 39.5 | 75 | 60.5 | |
| Possible income generation in MH | Disagree | 5 | 9.8 | 46 | 90.2 | <0.001 |
|  | Agree | 45 | 45.0 | 55 | 55.0 | |
| Psychiatric working environment | Unsatisfied | 18 | 18.2 | 81 | 81.8 | <0.001 |
|  | Satisfied | 32 | 61.5 | 20 | 38.5 | |
| Attitude towards MH and MI | Negative | 0 | 0.0 | 7 | 100.0 | 0.057 |
|  | Positive | 50 | 34.7 | 94 | 65.3 | |

Chi-square test

**Table 4. Factors associated with career preference in mental health (N = 151).**

| | $R^2$ | Career preference in mental health (Yes) | | p value |
|---|---|---|---|---|
| | | AOR | 95% CI | |
| Model A | | | | |
| Ever lived with a person with MI (ref: no) | 0.335 | 4.350 | 1.958, 9.664 | <0.001 |
| Model B | | | | |
| Ever been diagnosed/treated for MI (ref: no) | 0.233 | 0.999 | 0.307, 3.253 | 0.999 |
| Model C | | | | |
| Possible career advancement in MH (ref: disagree) | 0.333 | 16.193 | 2.022, 129.653 | 0.009 |
| Model D | | | | |
| Possible income generation in MH (ref: disagree) | 0.346 | 6.783 | 2.295, 20.047 | 0.001 |
| Model E | | | | |
| Satisfaction with psychiatric working environment (ref: unsatisfied) | 0.389 | 6.753 | 2.900, 15.726 | <0.001 |
| Model F | | | | |
| Attitude towards MH and MI (ref: negative) | 0.289 | $1.4 \times 10^9$ | 0.000, - | 0.999 |

Logistic regression analyses were performed. Sex, marital status, level of professional education, and religion were adjusted for in each model

Abbreviation: $R^2$, Nagelkerke's R squared; AOR, adjusted odds ratio; CI, confidence interval

with a person with mental illness was added in model A, being diagnosed or treated with a mental illness in model B, possibility of a career advancement in mental health in model C, possibility of income generation in mental health in model D, satisfaction with psychiatric working environment in model E, and attitude towards mental health and mental illness in model F. Among participants, living with a person with mental illness (adjusted odds ratio [AOR]: 4.350; 95% confidence interval [CI]: 1.958, 9.664; p <0.001), awareness of possible career advancement in mental health (AOR: 16.193; 95% CI: 2.022, 129.653; p = 0.009), awareness of possible income generation in mental health (AOR: 6.783; 95% CI: 2.295, 20.047; p = 0.001), and satisfaction with psychiatric working environment (AOR: 6.753; 95% CI: 2.900, 15.726; p <0.001), were associated with career preference in mental health. The Spearman's rank correlation coefficient ($\rho$) for the association between awareness of possible career advancement in mental health and possible income generation in the specialty was 0.398 (p<0.001).

## Discussion

This study was conducted to determine factors influencing career preference in mental health among nursing students and intern nurses in Dar es Salaam, Tanzania. The study showed low proportion of career preference in mental health among participants. Moreover, factors associated with career preference in mental health were revealed including ever-lived with a person with mental illness, awareness of possible career advancement in mental health, awareness of possible income generation in mental health, and satisfaction with psychiatric working environment. Other studies have reported differently the extent of career preference in mental health among nursing students and intern nurses and various factors influencing career choice in this area as discussed in the subsequent paragraphs [12, 15, 23–26].

Our study showed that only one third of participants preferred to have career preference in mental health nursing, implying limited number of future practitioners in this profession. Similarly, a much lower proportion of nursing students in Singapore (5.2%) and intern nurses in India (13%) considered mental health as a career choice [15, 23]. Career interest in mental health is not only very low among nursing students and intern nurses, but a similar situation is

observed in other professions in different parts of the world [12, 24–26]. On the contrary to our study, a greater proportion (two thirds) of nursing students in India intended to pursue their career as mental health nurses [27]. This shows that, depending on the context and other factors, mental health career preferences vary among nursing students and intern nurses, albeit in most cases, the preference is very low.

We found that having ever lived with a person with mental illness was associated with career preference in mental health. This corroborates with findings from other countries among medical students and interns [12, 28, 29]. This is further supported by the fact that a family history of psychiatric illness influences career preference in mental health [12]. It is argued that personal experience with mental illness may foster empathy and compassion that may influence career choice in mental health [30]. In a qualitative study, spending more time with patients and getting involvement in their psychosocial aspects of care was found to be appealing and influenced career choice in psychiatry [31]. It is also thought that the experience of knowing or caring for an individual with mental illness makes the student more comfortable with psychiatry and psychiatric patients. Moreover, first-hand students' experience of the shortcomings in the way psychiatric services are delivered, may inspire them to want to make a difference [28].

In the present study, awareness of possible career advancement and income generation in mental health were associated with career preference in mental health. This suggests that preference in the mental health profession is aligned with the financial reward. This finding is similar to the study among medical students in Australia which showed that students with prospects of a bright and interesting future in psychiatry and mental health have preference of this specialty [25]. This is also corroborated by a previous study in South Africa where lack of opportunities and career advancement were reported as barrier to career choice in mental health among undergraduate nursing students [32]. Lack of opportunities and career advancement in nursing practice is among commonly reported factors affecting nursing shortage [33]. This might be explained by participants in our study associating the awareness of possible career advancement in mental health with perception of possible income generation as indicated by a positive Spearman's rank correlation coefficient. This is further supported by previous studies in which students and interns who were highly interested in psychiatry ranked it as very attractive in respect to financial reward [23, 25]. Moreover, lack of awareness of possible career advancement in mental health is linked to students having anxiety surrounding mental illness due to stereotype, mostly negative perceptions of psychiatric patients and mental health care [34, 35].

The fact that psychiatric working environment in the current study influenced career preference in mental health, suggest that working condition is an essential attraction to the professionals. Similarly, this phenomenon is observed in other studies. Previous studies among undergraduate nursing students in South Africa and registered nurses in the Republic of Vanuatu showed that difficult working conditions such as heavy workload, lack of workforce, unusual working hours, and lack of support, were responsible for shortage of nurses in the psychiatric wards [32, 33]. Similarly, a study among medical students in Australia found difficult working environment in psychiatry as one of the contributing factor to later decision not to pursue training in psychiatry [17]. A qualitative study would provide more insight into specific issues related to work environment that affect career choice in mental health in our study setting.

Ever being diagnosed or treated for mental illness in this study was not associated with career preference in mental health. Studies assessing the association between having a history of mental illness and career choice or preference in mental health among students and interns are rare. Contrary to our results, one recent study demonstrated that having a mental health

condition is associated with a significantly greater likelihood of considering a career in mental health [14]. We could not find more a reliable scientific explanation for this except differences in sample size and socioeconomic conditions between the two studies. However, is it generally known that people with disabilities, including those with mental illness, experience career advancement challenges and reach career plateau [36].

Attitude towards mental health and mental illness in this study was not associated with career preference in mental health. Similarly, a systematic review regarding attitudes of undergraduate nursing students has consistently shown that mental health is one of the least preferred areas for a potential career [37, 38]. Stigma towards working with people with mental health difficulties remains prevalent and plays a significant role to negative attitudes toward career preference in mental health nursing [21, 31, 39]. A study among bachelor of nursing students in India emphasizes how negative stereotype toward people with mental illness significantly affects future career choices in psychiatric nursing [27]. Moreover, negative public attitude toward psychiatric nurses plays a role in lack of interest in working in psychiatric wards [40]. On the other hand, positive attitude towards psychiatry is associated with likelihood of choosing psychiatric nursing as a career [12]. To improve career interest in mental health, it is recommended that efforts should be made to improve the attitude of students toward this specialty, including measures to dispel misconceptions about psychiatry and sufficient counseling and informing students about mental health care [28, 35, 38].

Results of the current study have various implications to the nursing education and practice and mental health situation in the country. The low mental health career preference among nursing students and intern nurses is more likely to jeopardize the future of mental health nursing profession. Moreover, this situation may complicate the already existing challenge of the shortage of human resource for mental health in the country where the burden of mental disorders, like other low- and middle-income countries (LMICs), is very high [41]. If students are not interested in pursuing a career in mental health, efforts to close the existing mental health treatment gap will be hindered due to lack of access to trained healthcare providers [5–7, 42, 43].

There are a few circumstances limiting this study. First, it was conducted among nursing students and interns, hence it may not be generalizable to career preferences in mental health and psychiatry among other students in the health-related professions. Second, the study was conducted in a national tertiary hospital and university; it may not be representative of other training institutions and health facilities. Third, a relatively low sample size used may reduce the power of this study. Fourth, the models used in the multivariate analysis may not represent reality in an optimal way since the R-squared was low. However, we believe that results from this study may inform training institutions and healthcare providers to encourage students and interns to develop interest in mental health while addressing factors associated with mental health career preference.

## Conclusion

Low mental health career preference among college nursing students and intern nurses jeopardizes the future of the mental health nursing profession. It may complicate the already existing shortage of human resource for mental health and hinder efforts to close the huge treatment gaps that exists in LMICs. The overall results provide insights to stakeholders involved in training nurses (and other healthcare professionals) such as higher learning institutions, health facilities, and the Ministry of Health to take deliberate actions to ensure that interest to pursue a career in mental health is built among students and interns. Creation of awareness of possible career advancement and income generation in mental health among students and interns

needs to be emphasized, coupled with improving psychiatric working conditions/environment. We recommend a qualitative study to provide more insights into specific issues related to psychiatric working environment and how it affects career interest in mental health.

## Acknowledgments

We thank the Muhimbili University and Muhimbili National Hospital for granting permission to conduct this study. We also thank all nursing students and intern nurses for taking time to participate in this study and providing valuable information.

## Author Contributions

**Conceptualization:** Sofia Samson Sanga.

**Formal analysis:** Joel Seme Ambikile.

**Funding acquisition:** Sofia Samson Sanga.

**Investigation:** Sofia Samson Sanga.

**Methodology:** Edith A. M. Tarimo, Joel Seme Ambikile.

**Project administration:** Sofia Samson Sanga, Joel Seme Ambikile.

**Supervision:** Edith A. M. Tarimo.

**Validation:** Edith A. M. Tarimo.

**Writing – original draft:** Sofia Samson Sanga.

**Writing – review & editing:** Edith A. M. Tarimo, Joel Seme Ambikile.

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
