## [Decision Letter · Decision Letter 0]

3 Mar 2023

PGPH-D-22-01960

Factors Influencing Career Preference in Mental Health Among Nursing Students and Intern Nurses in Dar es Salaam, Tanzania

Dear Dr. Ambikile,

Thank you for submitting your manuscript to PLOS Global Public Health. After careful consideration, we feel that it has merit but does not fully meet PLOS Global Public Health’s publication criteria as it currently stands. Therefore, we invite you to submit a revised version of the manuscript that addresses the points raised by the reviewers during the review process.

We look forward to receiving your revised manuscript.

Kind regards,

Rakesh Singh

Academic Editor

Journal Requirements:

While revising the paper, please ensure you follow all the journal requirements.

Reviewers' comments:

Reviewer #1: Dear Authors, the manuscript is moderately interesting, needs major revisions. Following are some suggestions.

-Please use Mesh Terms as keywords.

-Please enter the period of the study in the abstract.

-Please describe the sampling better.

-Please in methods create the statistical analysis section better describing the analyzes carried out.

-Please check the formatting of the tables and the values inside.

-Please for logistic regression models show the pseudo-R2 (Pseudo-R-squared).

-Please in discussion better compare the results obtained with those present in the literature, the bibliography must be expanded.

-Please discuss in more detail the implications this study could have for public health, possibly create discussion in subchapter.

-Please improve the conclusions, right now they are quite trivial. Talk about how this study can be useful for the political decision-maker to evaluate the situation and possibly what corrective actions could be useful.

Reviewer #2: Factors Influencing Career Preference in Mental Health Among Nursing Students and Intern Nurses in Dar es Salaam, Tanzania

Thanks for the opportunity to review this paper; generally, the paper is well written, with a few recommendations for improvement.

Abstract

Indicate the sample type.

Do you mean recruited to complete the study questionnaires?

mentioned the instrument names.

AOR, adjusted odds ratio. Mention it completely for the first time.

Add an additional recommendation strategy.

Introduction

Recommend addressing and adding paragraphs about career choices, preferences, and influencing factors.

Try to support your introduction with a conceptual or theoretical framework.

and support your problem statement with previous or related studies.

There is a difference between your title and your purpose (career preference or career development), and both are different, so please be consistent.

Methods

 Please indicate

sociodemographic characteristics tool items

how many items the Career Preference in Mental Health and Related Factors tool has and how it scores them

Add recommendations for education, practice, and research.

---

## [Decision Letter · Decision Letter 1]

10 May 2023

PGPH-D-22-01960R1

Factors influencing career preference in mental mealth among nursing students and intern nurses in Dar es Salaam, Tanzania

Dear Dr. Ambikile,

Thank you for submitting your revised manuscript to PLOS Global Public Health. After careful consideration, we feel that it has merit but does not fully meet PLOS Global Public Health’s publication criteria as it currently stands. Therefore, we invite you to submit a revised version of the manuscript that addresses the points raised during the review process.

We look forward to receiving your revised manuscript.

Kind regards,

Rakesh Singh

Academic Editor

Journal Requirements:

1. Our staff editors have determined that your manuscript is likely within the scope of our Global Mental Health: challenges, opportunities, and the future of the field. This editorial initiative is headed by a team of Guest Editors for PLOS GPH: Rochelle Burgess (University College of London) and Dixon Chibanda (University of Zimbabwe and London School of Tropical Medicine and Hygiene). The Collection invites researchers to submit original research which engages with, or disrupts, the urgent needs across the global mental health landscape. We especially encourage submissions of studies that critically interrogate the status quo of the field and that involve inter-/trans-disciplinary approaches and those which share perspectives from underrepresented global regions and communities.

 Additional information can be found on our announcement page: https://collections.plos.org/call-for-papers/global-mental-health-opportunities-challenges/ 

If you would like your manuscript to be considered for this collection, please let us know in your cover letter and we will ensure that your paper is treated as if you were responding to this call.  Please note that being considered for the Collection does not require additional peer review beyond the journal’s standard process and will not delay the publication of your manuscript if it is accepted by PLOS GPH. If you would prefer to remove your manuscript from collection consideration, please specify this in the cover letter.

Reviewers' comments:

Reviewer #1: The authors have implemented the manuscript, it now needs minor revisions.

-Please check English and make the text more fluent.

-Please insert within the limits of the study the discourse that the proposed models do not represent reality in an optimal way, in fact the R-squared is rather low.

---

## [Editor Report · Decision Letter 2]

5 Jun 2023

Factors influencing career preference in mental mealth among nursing students and intern nurses in Dar es Salaam, Tanzania

PGPH-D-22-01960R2

Dear Dr. Ambikile,

We are pleased to inform you that your manuscript 'Factors influencing career preference in mental mealth among nursing students and intern nurses in Dar es Salaam, Tanzania' has been provisionally accepted for publication in PLOS Global Public Health.

Best regards,

Rakesh Singh

Academic Editor
